# Meta-analysis identifies five novel loci associated with endometriosis highlighting key genes involved in hormone metabolism

Yadav Sapkota[1,2], Valgerdur Steinthorsdottir[3], Andrew P. Morris[4,5], Amelie Fassbender[6,7], Nilufer Rahmioglu[5], Immaculata De Vivo[8,9], Julie E. Buring[8,10], Futao Zhang[11], Todd L. Edwards[12], Sarah Jones[13], Dorien O[6,7], Daniëlle Peterse[6,7], Kathryn M. Rexrode[8,10], Paul M. Ridker[8,10], Andrew J. Schork[14,15], Stuart MacGregor[1], Nicholas G. Martin[1], Christian M. Becker[16], Sosuke Adachi[17], Kosuke Yoshihara[17], Takayuki Enomoto[17], Atsushi Takahashi[18], Yoichiro Kamatani[18], Koichi Matsuda[19], Michiaki Kubo[18], Gudmar Thorleifsson[3], Reynir T. Geirsson[20,21], Unnur Thorsteinsdottir[3,21], Leanne M. Wallace[1,11], iPSYCH-SSI-Broad Group[†], Jian Yang[11], Digna R. Velez Edwards[22], Mette Nyegaard[23,24], Siew-Kee Low[18,*], Krina T. Zondervan[5,16,*], Stacey A. Missmer[8,9,*], Thomas D'Hooghe[6,7,25,*], Grant W. Montgomery[1,11,*], Daniel I. Chasman[8,10,*], Kari Stefansson[3,21,*], Joyce Y. Tung[26,*] & Dale R. Nyholt[1,27,*]

Endometriosis is a heritable hormone-dependent gynecological disorder, associated with severe pelvic pain and reduced fertility; however, its molecular mechanisms remain largely unknown. Here we perform a meta-analysis of 11 genome-wide association case-control data sets, totalling 17,045 endometriosis cases and 191,596 controls. In addition to replicating previously reported loci, we identify five novel loci significantly associated with endometriosis risk ($P < 5 \times 10^{-8}$), implicating genes involved in sex steroid hormone pathways (*FN1*, *CCDC170*, *ESR1*, *SYNE1* and *FSHB*). Conditional analysis identified five secondary association signals, including two at the *ESR1* locus, resulting in 19 independent single nucleotide polymorphisms (SNPs) robustly associated with endometriosis, which together explain up to 5.19% of variance in endometriosis. These results highlight novel variants in or near specific genes with important roles in sex steroid hormone signalling and function, and offer unique opportunities for more targeted functional research efforts.

[1] Department of Genetics and Computational Biology, QIMR Berghofer Medical Research Institute, Brisbane, Queensland 4006, Australia. [2] Department of Epidemiology and Cancer Control, St. Jude Children's Research Hospital, Memphis, Tennessee 38105, USA. [3] deCODE Genetics/Amgen, 101 Reykjavik, Iceland. [4] Department of Biostatistics, University of Liverpool, Liverpool L69 3GL, UK. [5] Wellcome Trust Centre for Human Genetics, University of Oxford, Oxford OX3 7BN, UK. [6] KULeuven, Department of Development and Regeneration, Organ systems, 3000 Leuven, Belgium. [7] Department of Obstetrics and Gynaecology, Leuven University Fertility Centre, University Hospital Leuven, 3000 Leuven, Belgium. [8] Harvard T.H. Chan School of Public Health, Boston, Massachusetts 02115, USA. [9] Channing Division of Network Medicine, Department of Medicine, Brigham and Women's Hospital and Harvard Medical School, Boston, Massachusetts 02115, USA. [10] Division of Preventive Medicine, Brigham and Women's Hospital, Boston, Massachusetts 02215, USA. [11] Institute for Molecular Bioscience, The University of Queensland, Brisbane, Queensland 4072, Australia. [12] Institute of Medicine and Public Health, Vanderbilt University Medical Center, Nashville, Tennessee 37203, USA. [13] Vanderbilt Genetics Institute, Division of Epidemiology, Institute of Medicine and Public Health, Department of Medicine, Vanderbilt University Medical Center, Nashville, Tennessee 37203, USA. [14] Cognitive Science Department, University of California, San Diego, La Jolla, California 92093, USA. [15] Institute of Biological Psychiatry, Mental Health Centre Sct. Hans, Copenhagen University Hospital, DK-2100 Copenhagen, Denmark. [16] Endometriosis CaRe Centre, Nuffield Dept of Obstetrics & Gynaecology, University of Oxford, John Radcliffe Hospital, Oxford OX3 9DU, UK. [17] Department of Obstetrics and Gynecology, Niigata University Graduate School of Medical and Dental Sciences, Niigata 950-2181, Japan. [18] Center for Integrative Medical Sciences, RIKEN, Yokohama 230-0045, Japan. [19] Institute of Medical Sciences, The University of Tokyo, Tokyo 108-8639, Japan. [20] Department of Obstetrics and Gynecology, Landspitali University Hospital, 101 Reykjavik, Iceland. [21] Faculty of Medicine, School of Health Sciences, University of Iceland, 101 Reykjavik, Iceland. [22] Vanderbilt Genetics Institute, Vanderbilt Epidemiology Center, Institute of Medicine and Public Health, Department of Obstetrics and Gynecology, Vanderbilt University Medical Center, Nashville, Tennessee 37203, USA. [23] Department of Biomedicine, Aarhus University, DK-8000 Aarhus, Denmark. [24] iPSYCH, The Lundbeck Foundation Initiative for Integrative Psychiatric Research, DK-2100 Copenhagen, Denmark. [25] Global Medical Affairs Fertility, Research and Development, Merck KGaA, Darmstadt, Germany. [26] 23andMe, Inc., 899 W. Evelyn Avenue, Mountain View, California 94041, USA. [27] Institute of Health and Biomedical Innovation, Queensland University of Technology, Queensland 4059, Australia. * These authors contributed equally to this work. Correspondence and requests for materials should be addressed to Y.S. (email: Yadav.Sapkota@stjude.org.) or to G.W.M. (email: g.montgomery1@uq.edu.au) or to D.R.N. (email: d.nyholt@qut.edu.au).
[†] A full list of consortium members appears at the end of the paper.

Endometriosis is a common gynecological disorder that affects 6–10% of women of reproductive age[1] and 20–50% of women with infertility[2]. The disease is associated with pelvic pain and is primarily characterized by the presence of endometrium-like tissue outside the uterus. The etiology of endometriosis is complex, involving multiple genetic and environmental risk factors. The condition has an estimated total heritability of 0.47–0.51 based on twin studies[1,3], and a common SNP-based heritability of 0.26 (ref. 4).

Genome-wide association (GWA) studies have identified 11 independent single-nucleotide polymorphisms (SNPs) for endometriosis. These SNPs include rs10965235 in *CDKN2BAS* on chromosome 9p21.3 identified in a Japanese ancestry GWA study[5]; rs1519761 on 2q23.3 identified in a US GWA study of European-ancestry women[6]; seven loci (rs7521902 near *WNT4* on 1p36.12, rs13391619 in *GREB1* on 2p25.1, rs4141819 on 2p14, rs7739264 near *ID4* on 6p22.3, rs12700667 on 7p15.2, rs1537377 near *CDKN2B-AS1* (independent of rs10965235) on 9p21.3 and rs10859871 near *VEZT* on 12q22) identified in a European-ancestry GWA study[7] and from a meta-analysis of European and Japanese ancestry GWA data[8]; and most recently rs17773813 near *KDR* on 4q12 and rs519664 in *TTC39B* on 9p22 in an Icelandic GWA study[9]. We also recently confirmed the suggested association of the *IL1A* gene locus on 2q13, by identifying genome-wide significant association between rs6542095 and endometriosis[10,11]. Thus, bringing the total to 12 independent SNPs associated with endometriosis at the genome-wide significance level, of which all but one (rs10965235 in *CDKN2BAS* on 9p21.3, identified in the Japanese GWA study[5]) are polymorphic in populations of European ancestry. Of the 11 European SNP risk loci, eight SNPs have been replicated and robustly implicated as susceptibility loci for endometriosis[10,12], the exceptions being rs1519761 (2q23.3), rs17773813 (4q12) and rs519664 (*TTC39B*) that are yet to be examined in an independent study.

To gain a better understanding of the genetic architecture of endometriosis, we sought to substantially expand upon the existing GWA data for endometriosis. Including 11 individual case-control data sets of European and Japanese ancestries (ten imputed using a recent 1000 Genomes Project reference panel and one directly genotyped), the current meta-analysis represents an approximate five-fold increase in the effective sample size[13] in comparison to the previously largest multi-ethnic GWA meta-analysis of 4,604 endometriosis cases and 9,393 controls from Australia, United Kingdom and Japan[8]. In addition to replicating 9 of the 11 previously reported European risk loci, this GWA meta-analysis identified 5 novel loci significantly associated with endometriosis risk ($P < 5 \times 10^{-8}$), implicating genes involved in sex steroid hormone pathways (*FN1*, *CCDC170*, *ESR1*, *SYNE1* and *FSHB*). Conditional analyses identified five novel secondary association signals at these implicated loci, including two at the *ESR1* locus, resulting in a total of 19 independent SNPs robustly associated with endometriosis, which together explain up to 5.19% of variance in endometriosis.

## Results

**Study overview.** This meta-analysis combined GWA data from 11 individual GWA case–control data sets, totalling 17,045 endometriosis cases and 191,858 controls of European and Japanese ancestries from Australia, Iceland, Belgium, the UK, the USA, Denmark and Japan. Individuals in the study are predominantly Europeans representing $\sim$93% of the total effective sample size (cases and controls), with the remaining $\sim$7% being of Japanese descent. A brief summary of the 11 individual GWA case–control data sets is provided in Supplementary Data 1. Seven

GWA data sets (QIMRHCS, OX, BBJ, deCODE, Adachi-6 and Adachi-500 K) have been published previously[5,8–10,14,15] and the remaining four are unpublished. Because individuals in the Adachi data set were genotyped using two different platforms (Affymetrix 500 K and 6.0), they were processed and analysed as separate GWA data sets. Details of individual data sets are provided in the Supplementary Information. All cases in the QIMRHCS, OX, deCODE and LEUVEN studies have surgically confirmed endometriosis and disease stage from surgical records using the revised American Fertility Society (rAFS) classification system[16]; women were grouped into Grade A (rAFS I or II disease or some ovarian disease with a few adhesions), Grade B (rAFS III or IV disease) or unknown stage as described previously[7]. Diagnosis of endometriosis in other studies is based on self-reports or a combination of surgical records and self-report (see Supplementary Information for more details).

Each GWA case–control data set followed similar quality control procedures and was imputed separately using the same 1000 Genomes Project reference panel (March 2012 Release), with the exception being the 23andMe and deCODE studies that were imputed using the 1000 Genomes Project October 2010 haplotypes and whole genome sequence data ($\sim$30 million sequence variants) of 8,453 Icelanders, respectively. The Adachi-6 data set consisted of only observed genotype data because individual-level genotypes were not available to carry out imputation. Genotypes (observed or imputed) in individual GWA case–control data sets were analysed and processed using similar approaches (see Methods section).

Primary GWA meta-analysis of all 17,045 endometriosis cases versus 191,858 controls, for the 6,979,035 SNPs that passed quality control in at least 50% (6 or more) of the participating studies, was performed using a fixed-effect model. SNPs with $P < 0.05$ in the primary GWA meta-analysis were further analysed after excluding cases with minimal or mild (Grade A) endometriosis (rAFS I or II disease) from the QIMRHCS, deCODE, LEUVEN and OX cohorts. A total of 391 SNPs reached conventional genome-wide significance ($P < 5 \times 10^{-8}$) and their association summary results are provided in Supplementary Data 2. SNPs showing suggestive association in the fixed-effect model and with evidence of heterogeneity ($P_{het} < 0.05$) were also analysed using the Han Eskin random-effects model (RE2)[17]. The RE2 model is similar to the traditional RE approach except it relaxes the conservative assumption in hypothesis testing and assumes no heterogeneity under the null hypothesis of no association. As such, it offers greater power under heterogeneity as compared with the conventional random-effects model. The genomic inflation factor ($\lambda$) for the GWA meta-analysis was 1.12. Quantile–quantile (Q–Q) plots for the fixed-effect GWA meta-analysis and 11 individual GWA case-control data sets are provided in Supplementary Figs 1–12.

**Risk loci associated with endometriosis.** Of the 11 previously reported European SNP risk loci, nine reached genome-wide significance ($P < 5 \times 10^{-8}$) in the present study (Fig. 1 and Table 1); we did not confirm associations at 2q23.3 and 9p22. The Q-Q plot for the fixed-effect GWA meta-analysis including all endometriosis cases ('All') after excluding the nine loci (the most significant ('index') SNPs and 1 Mb flanking region) is provided in Supplementary Fig. 13. SNP associations at loci on 1p36.12, 7p15.2 and 9p21.3 remained genome-wide significant when secondary meta-analysis ('Grade B') was performed after excluding cases with known minimal or mild (Grade A) endometriosis (rAFS I or II disease) (Table 1). Furthermore, all nine previously reported loci produced larger effects (odds ratios (ORs)) in Grade B analysis compared to the analysis of all cases—an observation

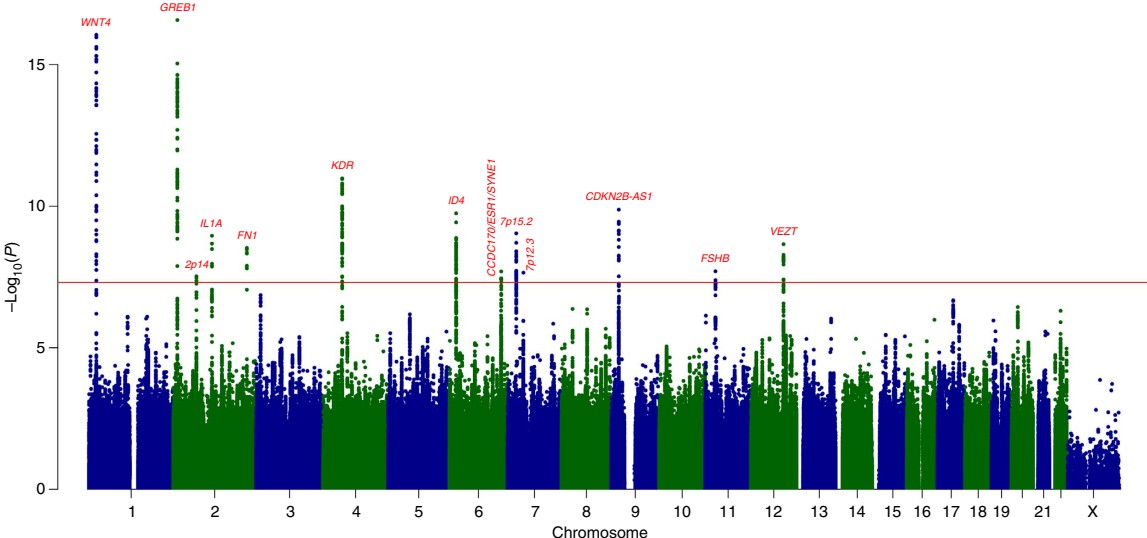

**Figure 1 | Manhattan plot for genome-wide associations with endometriosis.** Data are based on GWA meta-analysis of all endometriosis cases. The horizontal axis shows the chromosomal position, and the vertical axis shows the significance of tested markers combined in a fixed-effects meta-analysis. Markers that reached genome-wide significance ($P < 5 \times 10^{-8}$) are highlighted.

**Table 1 | Summary of the GWA meta-analysis results for 14 genome-wide significant loci.**

| Chr | SNP | Position (bp) | RA | OA | Meta-analysis (All) | | | | Meta-analysis (Grade B) | | Associated gene/cytoband |
|---|---|---|---|---|---|---|---|---|---|---|---|
| | | | | | $RAF_{EUR}$ | $RAF_{JPT}$ | OR (95% CI) | P value | OR (95% CI) | P value | |
| *Previously reported loci* | | | | | | | | | | | |
| 1 | rs12037376 | 22462111 | A | G | 0.17 | 0.58 | 1.16 (1.12–1.19) | $8.87 \times 10^{-17}$ | 1.28 (1.18–1.36) | $2.69 \times 10^{-9}$ | *WNT4/1p36.12* |
| 2 | rs11674184 | 11721535 | T | G | 0.61 | 0.54 | 1.13 (1.10–1.15) | $2.67 \times 10^{-17}$ | 1.18 (1.10–1.24) | $1.94 \times 10^{-6}$ | *GREB1/2p25.1* |
| 2 | rs6546324 | 67856490 | A | C | 0.31 | 0.21 | 1.08 (1.05–1.11) | $3.01 \times 10^{-8}$ | 1.19 (1.11–1.26) | $3.71 \times 10^{-7}$ | *ETAA1/2p14* |
| 2 | rs10167914 | 113563361 | G | A | 0.30 | 0.75 | 1.12 (1.08–1.15) | $1.10 \times 10^{-9}$ | 1.15 (1.07–1.21) | $7.59 \times 10^{-5}$ | *IL1A/2q13* |
| 4 | rs1903068 | 56008477 | A | G | 0.68 | 0.88 | 1.11 (1.07–1.13) | $1.04 \times 10^{-11}$ | 1.33 (1.24–1.40) | $2.58 \times 10^{-15}$ | *KDR/4q12* |
| 6 | rs760794 | 19790560 | T | C | 0.43 | 0.71 | 1.09 (1.06–1.12) | $1.79 \times 10^{-10}$ | 1.17 (1.10–1.24) | $8.74 \times 10^{-7}$ | *ID4/6p22.3* |
| 7 | rs12700667 | 25901639 | A | G | 0.74 | 0.20 | 1.10 (1.07–1.13) | $9.08 \times 10^{-10}$ | 1.28 (1.19–1.36) | $6.69 \times 10^{-11}$ | *7p15.2* |
| 9 | rs1537377 | 22169700 | C | T | 0.40 | 0.39 | 1.09 (1.06–1.12) | $1.33 \times 10^{-10}$ | 1.21 (1.13–1.27) | $6.31 \times 10^{-9}$ | *CDKN2B-AS1/9p21.3* |
| 12 | rs4762326 | 95668951 | T | C | 0.47 | 0.48 | 1.08 (1.05–1.11) | $2.20 \times 10^{-9}$ | 1.15 (1.08–1.21) | $1.08 \times 10^{-5}$ | *VEZT/12q22* |
| *Novel loci* | | | | | | | | | | | |
| 2 | rs1250241 | 216295312 | T | A | 0.29 | 0.06 | 1.06 (1.03–1.09) | $6.20 \times 10^{-5}$ | 1.23 (1.15–1.30) | $2.99 \times 10^{-9}$ | *FN1/2q35* |
| 6 | rs1971256 | 151816011 | C | T | 0.20 | 0.35 | 1.09 (1.06–1.13) | $3.74 \times 10^{-8}$ | 1.28 (1.19–1.36) | $1.50 \times 10^{-10}$ | *CCDC170/6q25.1* |
| 6 | rs71575922 | 152554014 | G | C | 0.16 | — | 1.11 (1.07–1.15) | $2.02 \times 10^{-8}$ | 1.35 (1.24–1.43) | $2.87 \times 10^{-12}$ | *SYNE1/6q25.1* |
| 7 | rs74491657 | 46947633 | G | A | 0.91 | 0.78 | 1.08 (1.03–1.13) | $1.23 \times 10^{-3}$ | 1.46 (1.28–1.59) | $2.24 \times 10^{-8}$ | *7p12.3* |
| 11 | rs74485684 | 30242287 | T | C | 0.84 | 0.98 | 1.11 (1.07–1.15) | $2.00 \times 10^{-8}$ | 1.26 (1.15–1.35) | $7.77 \times 10^{-7}$ | *FSHB/11p14.1* |

Chr, Chromosome; SNP, single-nucleotide polymorphism; Genomic position is shown relative to GRCh37 (hg19); GWA, genome-wide association; RA, risk allele; OA, other allele; OR, odds ratio with respect to RA; CI, confidence interval; $RAF_{EUR}$, average risk allele frequency in European samples; $RAF_{JPT}$, average risk allele frequency in Japanese samples.

consistent with previous reports of greater genetic loading in moderate-to-severe endometriosis. Regional association plots of the nine previously reported loci based on the fixed-effect meta-analysis results including all and Grade B endometriosis cases are provided in Supplementary Figs 14–22.

In addition to robustly implicating nine previously identified endometriosis SNP loci, the meta-analysis identified five new genomic regions harbouring risk loci for endometriosis (Fig. 1 and Table 1). In the 'All' fixed-effect meta-analysis, we observed genome-wide significant evidence for risk loci in *CCDC170* on 6q25.1 (rs1971256: OR (95% confidence interval (CI)) = 1.09 (1.06–1.13); $P_{all} = 3.74 \times 10^{-8}$), in *SYNE1* on 6q25.1 (rs71575922: OR (95% CI) = 1.11 (1.07–1.15); $P_{all} = 2.02 \times 10^{-8}$) and near *FSHB* on 11p14.1 (rs74485684: OR (95% CI) = 1.11 (1.07–1.15); $P_{all} = 2.00 \times 10^{-8}$). The 'Grade B' fixed-effect meta-analysis implicated rs1250241 in *FN1* on 2q35 (OR (95% CI) = 1.23 (1.15–1.30); $P_{Grade\ B} = 2.99 \times 10^{-9}$) and rs74491657 on 7p12.3

(OR (95% CI) = 1.46 (1.28–1.59); $P_{Grade\ B} = 2.24 \times 10^{-8}$) as genome-wide significantly associated with endometriosis risk (Table 1). Regional association plots of loci on 6q25.1 (*CCDC170* and *SYNE1*) and 11p14.1 (*FSHB*) based on fixed-effect meta-analysis results including all cases, and of loci on 2q35 and 7p12.3 based on results from fixed-effect meta-analysis for Grade B cases are provided in Fig. 2. Associations at the 6q25.1 *CCDC170* and *SYNE1* loci remained genome-wide significant in the 'Grade B' analysis, with larger effects (ORs) for the risk allele in comparison to those based on analysis of all endometriosis cases (Table 1). Regional association plots of the five newly identified loci based on fixed-effect meta-analysis including all and Grade B endometriosis cases are provided in Fig. 2 and Supplementary Fig. 23. Forest plots of risk allele effects (ORs) for the index SNPs at 14 loci in the individual GWA case-control data sets, and for the 'All' and 'Grade B' fixed-effect meta-analyses are given in Fig. 3 and Supplementary Figs 24–33.

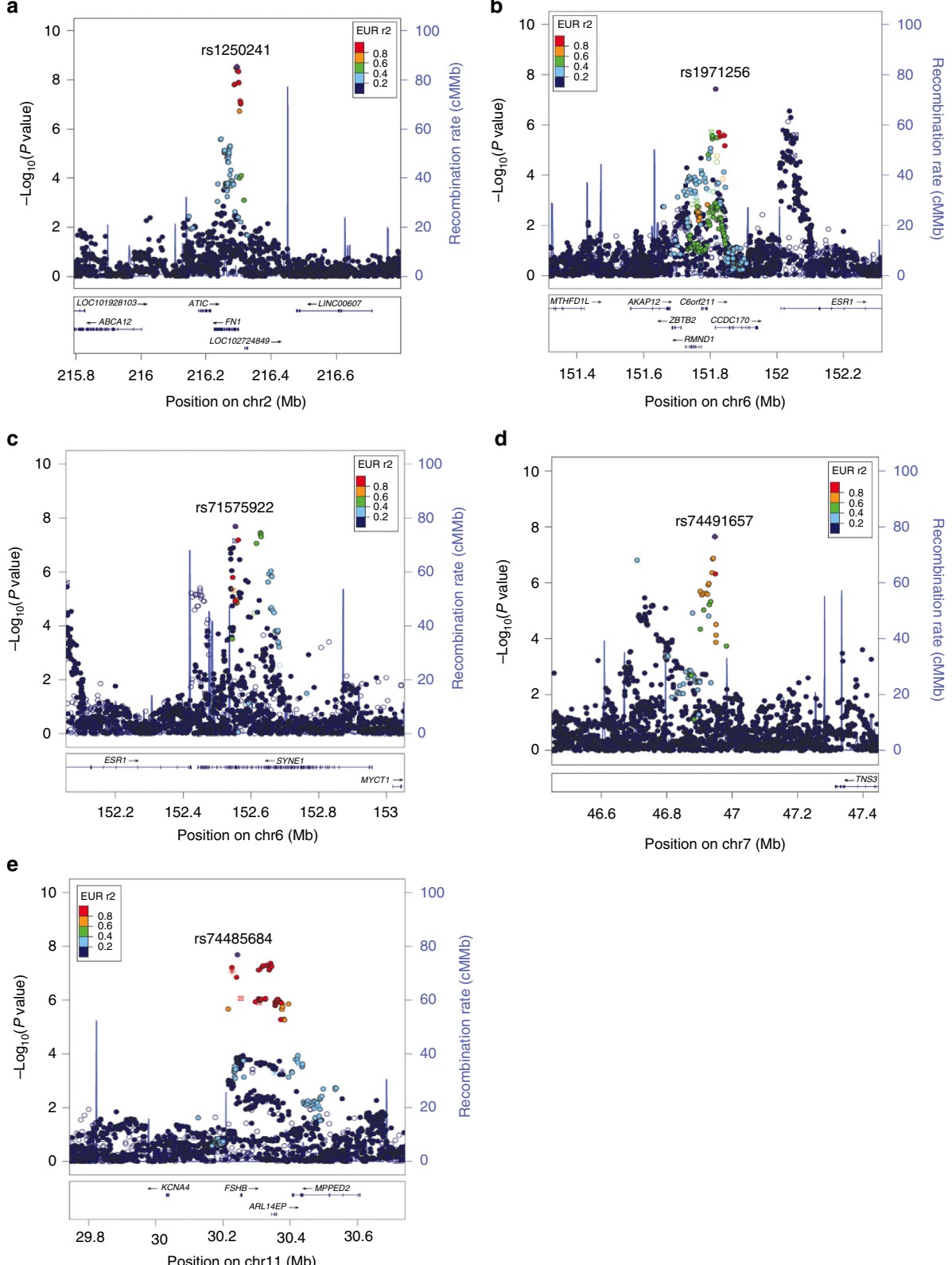

**Figure 2 | LocusZoom plots of five genome-wide significant endometriosis loci.** Association with endometriosis is expressed as $-\log_{10}(P$ value) for five new genome-wide significant loci: *FN1* 2q35 (2a), *CCDC170* on 6q25.1 (2b), *SYNE1* on 6q25.1 (2c), 7p12.3 (2d), and near *FSHB* on 11p14.1 (2e). Results for 2q35 and 7p12.3 are based on analysis including only moderate-to-severe ('Grade B') endometriosis cases. SNPs are shown as circles, diamonds or squares (filled or unfilled), with the top SNP represented by purple colour. All other SNPs are colour coded according to the strength of LD with the top SNP (as measured by $r^2$ in the European 1000 Genomes data).

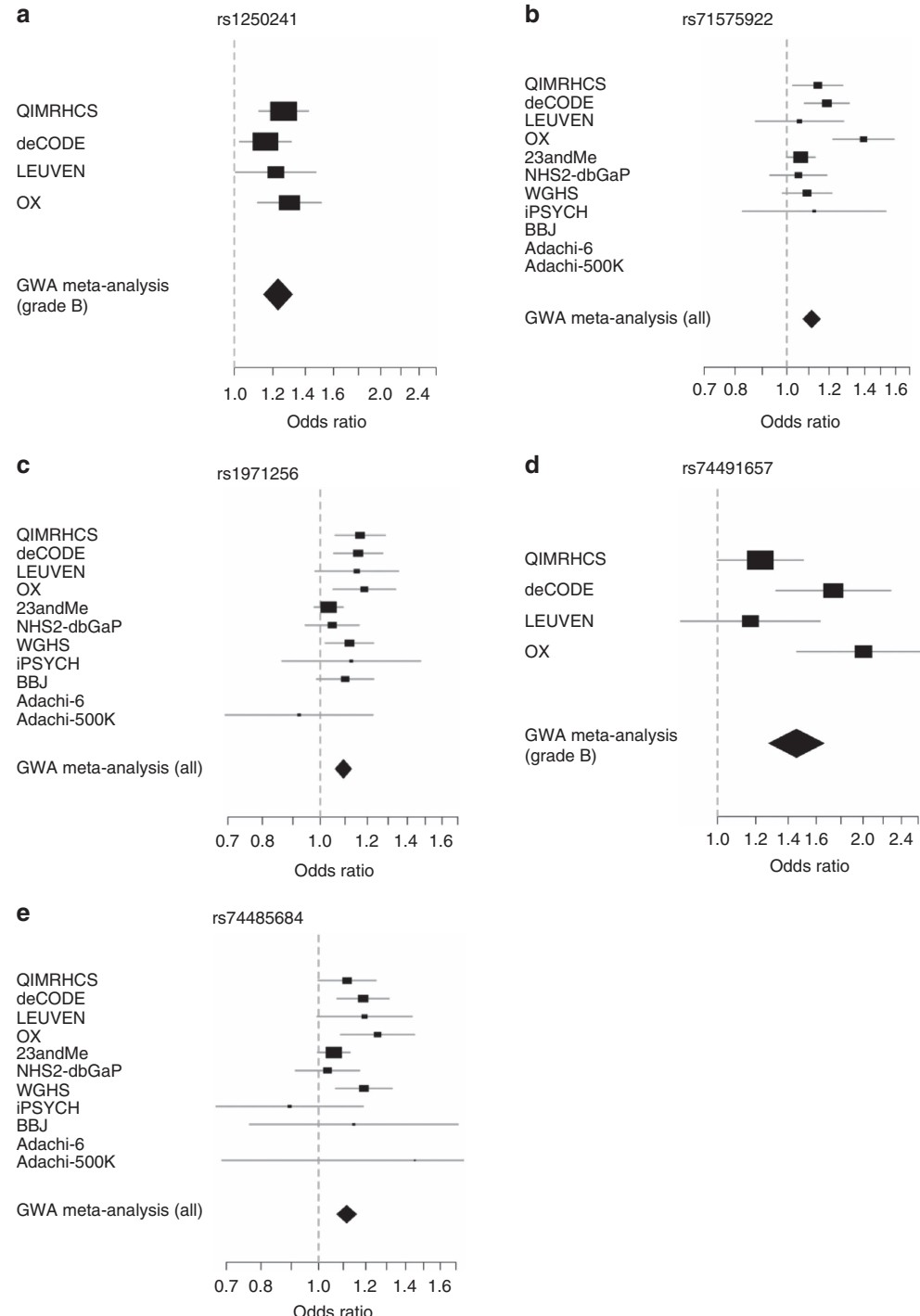

**Figure 3 | Forest plots showing risk allele effects for five endometriosis loci.** Risk allele effects for the five new genome-wide significant loci in the individual case-control data sets and GWA meta-analysis: *FN1* 2q35 (3a), *CCDC170* on 6q25.1 (3b), *SYNE1* on 6q25.1 (3c), 7p12.3 (3d), and near *FSHB* on 11p14.1 (3e). Results for 2q35 and 7p12.3 are based on analysis including only moderate-to-severe ('Grade B') endometriosis cases. Risk allele effects of the remaining three SNPs are from analysis including all endometriosis cases.

**Distinct association signals at endometriosis risk loci.** To identify distinct secondary association signals at the 14 loci, we used the genome-wide complex trait analysis (GCTA) software[18] to perform approximate conditional analysis based on summary statistics from meta-analysis including all endometriosis cases. For the *FN1* 2q35 and 7p12.3 loci, we used summary statistics from meta-analysis including Grade B endometriosis cases. We conservatively defined a locus as the chromosomal region 500 kb up- and down-stream of the index SNP at the locus.

We estimated the effective number of independent SNPs to be 11,631 across all 14 regions. We therefore used a region-wide Bonferroni adjusted significant threshold of $P < 4.3 \times 10^{-6}$ to declare a secondary association signal if a SNP achieved this threshold after conditioning on the index SNP at each locus. GCTA identified five secondary signals including one (rs77294520 near *GREB1*) on 2p25.1; two (rs2206949 in *ESR1* and rs17803970 in *SYNE1*) on 6q25.1, and two (rs10757272 in *CDKN2B-AS1* and rs1448792) on 9p21.3 (Table 2). Of these,

rs77294520 on 2p25.1 also remained significant in Grade B analyses with a larger effect (OR). Results for all SNPs with $P < 4.3 \times 10^{-6}$ in the GCTA conditional analysis based on summary statistics including all studies are provided in Supplementary Data 3. We also performed additional conditional analysis using European samples alone. Except for near region-wide significance for rs17803970 on 6q25.1 (*SYNE1*; $P = 4.59 \times 10^{-6}$) and for rs10757272 on 9p21.3 (*CDKN2B-AS1*; $P = 1.40 \times 10^{-5}$), the remaining three secondary association signals persisted with region-wide significance when analysis was restricted to Europeans studies only (Supplementary Data 4). Furthermore, there was no linkage disequilibrium (LD) or very low LD ($r^2 < 0.03$) between the index SNPs and the five secondary association signals on 2p25.1, 6q25.1 and 9p21.3 in both European and Japanese populations (Supplementary Data 5), suggesting that the results are not influenced by differences in LD patterns across European and Japanese populations. Regional association and forest plots for the five secondary signals are provided in Supplementary Figs 34–42. Taken together, these data implicate 19 independent SNPs at 14 distinct genomic loci, including four on 6q25.1—a locus containing a cluster of genes including *ESR1*.

On the basis of the fixed-effect GWA meta-analysis results including all endometriosis cases from European studies, the nine previously reported SNPs explain 0.97% of the phenotypic variance on the liability scale[19]. The 10 new SNPs identified in this study together explain a further 0.78%, totalling to 1.75% of the phenotypic variance explained for endometriosis. More importantly, the 19 independent SNPs together explained 5.19% of the phenotypic variance in Grade B endometriosis cases, of which 2.46% was explained by the 10 new SNPs.

**SNP effects based on ancestry and endometriosis definition.** Of the 14 distinct loci, 6 (2p14, 2q35, 4q12, 6q25.1 (*SYNE1*), 7p15.2 and 12q22) showed evidence of between-study heterogeneity ($P_{het} < 0.05$; $I^2$: 13.30–20.33) in fixed-effect meta-analysis including all endometriosis cases; however, after appropriately modelling the observed heterogeneity in the RE2 model, these associations remained genome-wide significant (Supplementary Data 2).

None of the six loci showed between-study heterogeneity in analyses restricted to Japanese alone and self-reported endometriosis cases (Supplementary Data 6). In Europeans, five loci (2p14, 2q35, 4q12, 6q25.1 (*SYNE1*) and 7p15.2) showed heterogeneity in allelic associations, and except for 6q25.1 (*SYNE1*), this trend persisted in surgically confirmed endometriosis cases. However, except for 2p14 and 4q12, this heterogeneity attenuated in moderate-to-severe endometriosis cases (Supplementary Data 2). All five loci except 2p14 produced larger effect sizes for surgically confirmed endometriosis than diagnosis based on self-reports. With respect to 2p14, residual between-study heterogeneity may be driven by opposite direction of effect in two out of the eight European studies (Supplementary Data 2 and Supplementary Fig. 26).

Most SNPs showed larger effect sizes in Japanese populations in comparison with results from Europeans alone (Supplementary Data 6). Whereas all 19 SNPs showed significant associations with endometriosis ($P < 2.6 \times 10^{-3}$ after multiple testing) in Europeans, only six SNPs showed significance in Japanese alone, although importantly, direction of effects for all SNPs were concordant with those produced in the meta-analysis. For surgically confirmed endometriosis, all but two SNPs (rs2206949 on 6q25.1 and rs10757272 on 9p21.3) showed significant association with endometriosis ($P < 2.6 \times 10^{-3}$ after multiple testing).

**Endometriosis risk SNPs associated with other traits.** We checked genome-wide significant associations of the 19 SNPs with other diseases or traits using the NHGRI GWAS catalogue[20]. We searched for 395 SNPs including the 19 SNPs as well as all other SNPs in high LD ($r^2 > 0.7$) with the 19 SNPs (Supplementary Data 7). Of these, we observed associations with multiple diseases or traits, including epithelial ovarian cancer[21], low-density lipoprotein cholesterol[22], coronary heart disease[23], luteinizing hormone and follicle-stimulating hormone levels, and age at onset for menopause[24].

We did not observe statistically significant genetic correlation between endometriosis and 199 common complex traits, based on LD score regression analysis[25] using LD hub[26]. Genetic correlations with nominal $P < 0.05$ are provided in Supplementary Figs 43,44.

Considering that SNPs at 6q25.1 are reported to be associated with breast cancer and related phenotypes, we investigated for overlap of association signals between breast cancer and endometriosis. A recent study based on the custom-designed iCOGS data in 118,816 women reported evidence for at least 5 independent risk variants at 6q25.1, each associated with different breast cancer phenotypes, including oestrogen receptor, human ERBB2 tumour subtypes, mammographic density, and tumour grade[27]. We found no overlap (LD $r^2 < 0.17$) between these five breast cancer signals and our four independent SNPs at the 6q25.1 (*ESR1*) locus. We therefore obtained association summary results of 101 SNPs with genome-wide significance for overall breast cancer from Dunning *et al.*[27], and cross-checked with our GWA meta-analysis results (Supplementary Data 9). Of these, 23 SNPs also showed associations ($P < 0.05$) with endometriosis. The risk allele of six SNPs was the same for both breast cancer and endometriosis, including four SNPs (rs851981, rs851980, rs2206948 and rs150182883) that are in strong LD ($r^2 \geq 0.60$) with our secondary association signal rs2206949 at 6q25.1 (*ESR1*). The secondary endometriosis risk SNP rs2206949 was also strongly associated with overall breast cancer ($P = 5.5 \times 10^{-6}$). Based on the 1000 Genomes Project European reference data, the 101 and 23 SNP-sets correspond to four and two independent SNPs, respectively, thereby suggesting significant genetic overlap between overall breast cancer and endometriosis ($P = 0.02$; binomial test) at the 6q25.1 (*ESR1*) locus.

**Genes associated with endometriosis.** A genome-wide gene-based test using VEGAS2 (ref. 28) identified 18 genes that reached our conservative gene-based threshold of $P < 2.23 \times 10^{-6}$ (Supplementary Data 10); we also provide results for all genes with combined $P < 0.05$ for reference. Of these, 12 genes are located at six GWA SNP risk loci including 1p36.12 (*WNT4*, *LINC00339*, *LOC101928043* and *CDC42*), 2p25.1 (*GREB1*), 2q13 (*IL1A* and *CKAP2L*), 7p15.2 (*RNU6-16P*), 9p21.3 (*CDKN2A*) and 12q22 (*MIR331*, *MIR3685* and *VEZT*). Notably, the remaining six gene-based genome-wide significant association signals were at three novel genomic regions 1q24.3 (*DNM3OS*, *MIR214* and *MIR3120*), 9q22.32 (*MIR23B*, *MIR27B*) and 16p13.3 (*LINC00921*).

**Fine-mapping of endometriosis risk loci.** To identify potential causal variants responsible for the 19 independent association signals, we performed fine-mapping analysis based on our GWA meta-analysis results including all studies except for the Adachi-6 data set, as well as using results from only Europeans. (Supplementary Data 11). We assumed a single causal variant for each association signal and constructed a 99% credible set of variants including SNPs within 500 kb of the index SNP. Except for the 6q25.1 (*SYNE1*) locus, the length of the 99% credible

**Table 2 | Secondary association signals based on summary statistics including all studies and the combined QIMRHCS and 1000G JPT samples to calculate LD and corresponding results using summary statistics from Grade B analysis and QIMRHCS for LD estimation.**

| Chr | SNP | Position (bp) | RA | OA | RAF$_{EUR}$ | RAF$_{JPT}$ | Endometriosis cases | RAF | OR (95% CI) | P value | Freq_ref | OR$_{cond}$ (95% CI) | P$_{cond}$ | Associated gene/ cytoband |
|---|---|---|---|---|---|---|---|---|---|---|---|---|---|---|
| 2 | rs77294520 | 11660955 | C | G | 0.147 | — | All | 0.147 | 1.16 (1.11–1.21) | $9.91 \times 10^{-13}$ | 0.144 | 1.13 (1.09–1.18) | $1.67 \times 10^{-9}$ | GREB1/2p25.1 |
|   |   |   |   |   |   |   | Grade B | 0.150 | 1.29 (1.18–1.42) | $1.45 \times 10^{-8}$ | 0.147 | 1.26 (1.15–1.37) | $4.76 \times 10^{-7}$ |   |
| 6 | rs2206949 | 152037556 | T | C | 0.270 | 0.162 | All | 0.262 | 1.10 (1.06–1.14) | $2.73 \times 10^{-7}$ | 0.284 | 1.11 (1.07–1.15) | $2.68 \times 10^{-8}$ | ESR1/6q25.1 |
|   |   |   |   |   |   |   | Grade B | 0.275 | 1.09 (1.01–1.17) | $2.49 \times 10^{-2}$ | 0.286 | 1.11 (1.03–1.19) | $5.19 \times 10^{-3}$ |   |
| 6 | rs17803970 | 152553718 | A | T | 0.918 | 0.970 | All | 0.920 | 1.15 (1.09–1.21) | $7.04 \times 10^{-8}$ | 0.919 | 1.13 (1.07–1.18) | $3.25 \times 10^{-6}$ | SYNE1/6q25.1 |
|   |   |   |   |   |   |   | Grade B | 0.921 | 1.35 (1.18–1.53) | $4.75 \times 10^{-6}$ | 0.918 | 1.27 (1.12–1.44) | $2.21 \times 10^{-4}$ |   |
| 9 | rs10757272 | 22088260 | C | T | 0.521 | 0.383 | All | 0.504 | 1.07 (1.04–1.10) | $2.60 \times 10^{-7}$ | 0.506 | 1.07 (1.04–1.10) | $6.35 \times 10^{-7}$ | CDKN2B-AS1/ 9p21.3 |
|   |   |   |   |   |   |   | Grade B | 0.524 | 1.09 (1.02–1.16) | $1.12 \times 10^{-2}$ | 0.511 | 1.08 (1.01–1.15) | $1.71 \times 10^{-2}$ |   |
| 9 | rs1448792 | 22641633 | G | A | 0.757 | 0.661 | All | 0.741 | 1.08 (1.05–1.12) | $1.79 \times 10^{-7}$ | 0.751 | 1.08 (1.05–1.11) | $7.03 \times 10^{-7}$ | 9p21.3 |
|   |   |   |   |   |   |   | Grade B | 0.759 | 1.06 (0.98–1.14) | $1.21 \times 10^{-1}$ | 0.753 | 1.05 (0.97–1.13) | $1.99 \times 10^{-1}$ |   |

Chr, Chromosome; SNP, single-nucleotide polymorphism; Genomic position is shown related to GRCh37 (hg19); Freq_ref, frequency of the risk allele in the reference sample; LD, linkage disequilibrium; RA, risk allele; OA, other allele; RAF, OR (95% CI) and P, risk allele frequency, odds ratio and 95% confidence interval, and P value from the meta-analysis; OR$_{cond}$ (95% CI) and P$_{cond}$, odds ratio and 95% confidence interval, and P value from conditional analyses; RAF$_{EUR}$, risk allele frequency in Europeans; RAF$_{JPT}$, risk allele frequency in Japanese;.

interval and the number of credible SNPs for all association signals were smaller in analysis including all studies compared to the results restricted to only Europeans. Based on results from all studies, the smallest 99% credible interval was ∼16.52 kb observed for rs11674184 on 2p25.1 and the largest was ∼604.80 kb for the 6q25.1 (SYNE1). The 99% credible sets for the 19 independent associations using GWA meta-analysis results including all studies and only Europeans are provided in Supplementary Data 12.

**Bioinformatic analyses of endometriosis risk loci.** We then examined the *cis* associations between the 19 independent SNPs and other SNPs in high LD ($r^2 > 0.7$) with the lead SNPs (Supplementary Data 7), and expression of nearby genes in whole blood, breast, cervix, muscle, ovary, uterus and adipose tissues using the GTEx eQTL portal[29]. Of these, the most relevant tissue for endometriosis is a small set of 32 uterine samples which are a mixture of both endometrium and myometrium. We found strong significant associations (false discovery rate (FDR) < 0.05) between a SNP and expression of nearby genes in subcutaneous adipose tissues (Supplementary Data 13). Risk allele (G) of rs56376645 on 7p15.2 which is associated with endometriosis at genome-wide significance (OR (95% CI) = 1.09 (1.06–1.12); $P_{all} = 1.93 \times 10^{-8}$) also showed strong associations (beta = 0.38; $P = 1.44 \times 10^{-7}$; FDR = $1.28 \times 10^{-3}$) with increased expression of AC003090.1 (Supplementary Fig. 45).

To identify potentially causal genes underlying the identified endometriosis associations, we used a novel method[30], summary data-based Mendelian randomization (SMR), which exploits the concept of Mendelian Randomisation (MR), to test for the causative effect of an exposure (that is, gene expression) on a phenotypic outcome (that is, endometriosis) using a genetic (SNP) variant as an instrumental variable. We used the method to identify causal genes at our endometriosis risk loci, using the GWA meta-analysis summary results from all studies including all endometriosis cases, and the eQTL summary data from Westra *et al.*[31], an eQTL meta-analysis of 5,311 samples from peripheral blood, with SNPs imputed to the HapMap2 reference panel. The SMR analysis identified two potential causal genes, *CDC42* (rs2268177; $P_{SMR} = 1.07 \times 10^{-12}$) and *VEZT* (rs14121; $P_{SMR} = 3.41 \times 10^{-6}$), underlying endometriosis loci at 1p36.12 and 12q22, respectively (Supplementary Data 14). For rs2268177, there was no significant evidence of heterogeneity ($P_{HEIDI} = 0.065$) in effect sizes of dependent SNPs at this associated region and therefore it is very likely that rs2268177 contributes to both endometriosis risk and the expression level of *CDC42*. Significant heterogeneity ($P_{HEIDI} < 0.05$) may indicate the

possibility of two causal variants at a locus: one affecting endometriosis risk and the other affecting expression level of the gene (Supplementary Data 15). Significant heterogeneity ($P_{HEIDI} = 0.004$) for rs14,121 was observed, suggesting that the observed causal effect of *VEZT* expression on risk of endometriosis may be due to colocalization, but this needs to be investigated further.

DEPICT[32] analysis provided little support for potential genes, pathways, tissues or cell types reaching multiple testing threshold (FDR ≤ 0.05). However, when using results with evidence of genome-wide suggestive association ($P < 1 \times 10^{-5}$) in either all or Grade B meta-analysis, DEPICT provided evidence for significant enrichment (FDR ≤ 0.05) of COPB1 PPI subnetwork gene set (Supplementary Data 16). Additionally, suggestive evidence for enrichment (FDR ≤ 0.2) was observed for ten tissues, including female genitalia, uterus, endocrine glands, endometrium, ovary and Fallopian tubes (Supplementary Data 16).

## Discussion

We conducted a GWA meta-analysis of ∼7 million SNPs in 17,045 endometriosis cases and 191,596 controls, confirmed 9 out of 11 previously reported European risk loci, and identified five new genomic regions in or near *CCDC170*, *SYNE1*, *FSHB*, *FN1* and 7p12.3 harbouring endometriosis risk loci. This study represents a nearly fivefold increase in sample size in comparison with the previously largest endometriosis discovery GWA study and provided evidence for five secondary association signals including *ESR1*. The variance explained by the ten newly identified SNPs in all and Grade B cases was 0.78% and 2.46%, bringing the total variance explained for endometriosis to 1.75% and 5.19%, respectively, when considering all 19 associated SNPs. Importantly, our results highlight key genes involved in hormone metabolism that are likely to play a major role in endometriosis pathogenesis, thereby advancing current knowledge of endometriosis biology.

Previous GWA studies of endometriosis have implicated WNT signalling, oestrogen responsive genes and genes involved in the actin cytoskeleton and cellular adhesion[12,33]. Target genes in most regions are yet to be identified, but there is evidence to support candidates in several regions as previously described[34]. The most strongly associated (index) SNP for endometriosis at the *WNT4* locus on chromosome 1p36.12 is also the index SNP for ovarian cancer[35] and the risk mechanism likely acts through inverse regulation of *CDC42* and *LINC00339* (ref. 36). The index SNP associated with endometriosis on chromosome 2p25.1 is a common splice variant in the oestrogen-responsive growth

regulation by oestrogen in breast cancer 1 (*GREB1*) gene[33,37] and SNPs associated with endometriosis on chromosome 12q22 increase expression of the transmembrane adherens junctions protein coding gene vezatin (*VEZT*) in RNA from blood and endometrium[38]. We confirm association near the kinase insert domain receptor (*KDR*) gene that was recently reported by the Icelandic GWA study[9]. The gene encodes vascular endothelial growth factor receptor 2, which promotes proliferation, survival, migration and differentiation of endothelial cells. *KDR* is responsive to steroid hormones; expression of *KDR* in endometrial stromal cells isolated from proliferative phase endometrium was significantly increased by oestrogen and medroxyprogesterone acetate *in vitro*[39]. During the premenstrual phase in both humans and macaques, *KDR* expression was significantly increased in the stromal cells of the endometrium[40]. Results from this meta-analysis support and extend these observations, but further functional studies are needed to identify and confirm the causal genes in all regions.

We identified several independent signals in a region that includes *ESR1* encoding oestrogen receptor 1 on chromosome 6p25.1. Endometriosis is an oestrogen-dependent disease. Symptoms occur after puberty and oestrogen action contributes to pathological processes including growth of lesions and inflammation, and to the symptoms including pain[41,42]. Our primary meta-analysis identified two SNPs at this locus—rs1971256 in *CCDC170* and rs71575922 in *SYNE1*—located up and downstream of *ESR1*, respectively. Conditional analysis identified a further two independent associations at this locus, including rs2206949 in *ESR1* and rs17803970 in *SYNE1*. The 6q25.1 region is a well-established susceptibility locus for breast cancer in both Europeans and Asians[27,43,44]. Variants at this locus have recently been associated with overall breast cancer and its sub-phenotypes including tumour subtypes, mammographic density and tumour grade[27]. Of the four independent endometriosis SNPs at this locus, the association signal for rs2206949 in intron 2 of *ESR1* overlaps with the signal observed for overall breast cancer. In fact, we observed a significant ($P = 0.02$; binomial test) genetic overlap (sharing) between genetic risk for overall breast cancer and endometriosis at this locus, highlighting the importance of hormonal influences of both diseases which warrants further detailed exploration.

Our results support a role for a functional effect associated with endometriosis on chromosome 11p14.1. The association signal includes an LD region beginning upstream of *FSHB* and extending across to *ARL14EP*. The strongest signal was for SNP rs74485684 (risk allele T, RAF = 0.84, OR = 1.11) located 10,276 bp upstream of the transcription start site of *FSHB*. Nominal evidence for association between endometriosis and SNPs upstream of *FSHB* was recently reported in independent samples from the UK Biobank providing strong support for this result[45]. *FSHB* encodes the beta polypeptide of FSH, a glycoprotein hormone that plays a central role in ovarian folliculogenesis[46]. Our index SNP rs74485684 is in high LD with other SNPs located in this region upstream of *FSHB* including rs11031005 ($r^2 = 0.82$) associated with FSH concentrations and rs11031002 ($r^2 = 0.64$) associated with LH concentrations[24]. FSH and LH are related gonadotropin hormones sharing a common alpha subunit. The LH beta subunit is located on chromosome 19q13.32 and association between these SNPs on chromosome 11 with concentrations for both hormones suggests a common mechanism of regulation. They both play central roles in regulating follicle development in the ovary, influencing oestradiol release during the proliferative phase of the cycle[46] and contributing to a role for estradiol in endometriosis risk. The allele(s) associated with increased risk of endometriosis are also associated with shorter menstrual cycles, earlier age at menopause, increased dizygotic twinning and polycystic ovarian syndrome[24,45,47,48].

Our results provide further support for association at the 2p25.1 locus, containing an oestrogen-regulated gene, *GREB1*, that was first identified in breast cancer cell lines and tumours[49]. In addition to confirming the previously identified association signal at this locus, our results provide evidence for secondary association with risk of endometriosis. Regulation of *GREB1* transcription by oestrogen receptor α (ERα) is mediated through three oestrogen response elements (EREs) located 20 kb upstream of the gene[50,51]. Moreover, GREB1 functions as an essential component of the oestrogen receptor transcription complex[52], and while effects of individual risk SNPs are small, the results suggest that risk variants acting on several genes in the same pathway act to increase sensitivity to oestrogen and increase the risk of endometriosis.

This study robustly associated the *FN1* locus with endometriosis, in particular with moderate-to-severe disease. Association between rs1250248, which is in very high LD ($r^2 = 0.95$) with our lead SNP rs1250241 at this locus, was first reported by the earlier European GWA study led by us, but was not replicated in an independent sample[7]. Results from the current study provide genome-wide evidence for the *FN1* locus in Grade B endometriosis, thereby highlighting the importance of subgroup analysis and phenotype definition. *FN1* encodes fibronectin, which is a glycoprotein of the extracellular matrix and is also present in plasma, and at the cell surface. Fibronectin is involved in important cellular activities including cell adhesion, growth and migration, and it also plays a critical role in would healing, blood coagulation and metastasis[53].

Our results provide strong evidence for multiple association signals on chromosome 9p21.3 containing *ANRIL* (antisense non-coding RNA in the *INK4* locus, also known as *CDKN2B-AS1*). A recent Japanese study[54] showed allele specific effects of rs17761446 on regulation of ANRIL expression with twofold greater chromatin interactions for the protective G allele with the *ANRIL* promoter. SNP rs17761446 is monomorphic in Europeans and is also in weak LD ($r^2 < 0.16$) in Japanese with the SNPs implicated in the current study, suggesting that these associations are independent endometriosis-specific risk loci. SNPs at the chromosome 9p21 locus are also associated with a number of other diseases, including coronary artery disease (CAD)—a disease in which *ANRIL* has previously been implicated[55]. Chromatin conformation capture in this region in human vascular endothelial cells identified short-range interactions between sequences at the 9p21.3 locus and sequences in the vicinity of the genes encoding *CDKN2A*, *CDKN2B*, and *MTAP*, and long-range interactions with *IFNW1* and *IFNA21* approximately one million base pairs upstream on chromosome 9 (ref. 56). Functional studies in mammalian cells show that ANRIL overexpression accelerated proliferation, increased adhesion and decreased apoptosis[54,56]. These functions may be important in endometriosis and hence, additional studies will be necessary to understand how SNPs at this locus in both Japanese and European populations influence endometriosis.

The majority of the identified SNP loci showed larger effects in the Grade B analysis in comparison with those based on analysis of all endometriosis, supporting previous findings of greater genetic loading in moderate-to-severe endometriosis[7,8,57]. The smaller effect sizes in analyses including all endometriosis cases may be due to disease misclassification in self-reported endometriosis cases. However, this would only be part of the explanation. We have previously shown that the genetic contribution to phenotypes of surgically confirmed endometriosis or minimal disease is weaker than for severe disease phenotypes[7,57]. Using polygenic prediction analysis, we also

showed significant prediction of minimal disease between two independent data sets with surgically confirmed endometriosis[57].

In summary, this GWA meta-analysis of endometriosis provides evidence for 10 new independent SNP loci, and more than doubles the proportion of genetic variation in endometriosis explained by robust SNP risk loci. These results identify novel variants in or near specific genes with important roles in sex steroid hormone signalling and function, and offer unique opportunities for more targeted functional research efforts.

## Methods

**Study overview and GWA Genotyping.** Our study included 17,045 endometriosis cases and 191,596 controls from 11 individual case–control data sets of European and Japanese ancestry. The European ancestry arm included 14,926 cases and 189,715 controls from eight individual case–control data sets and the Japanese data sets included 2,119 cases and 2,143 controls from three cohorts. The samples were genotyped on a variety of commercial arrays, as outlined in the Supplementary Information. All samples were collected with informed consent and study protocols were approved by the relevant local institutional ethics committees: the QIMR Human Research Ethics Committee (QIMR), the University of Newcastle and Hunter New England Population Health Human Research Ethics Committees (HCS), the Oxford regional multi-center and local Research Ethics Committees (UK), the Ethical Committees at the Institute of Medical Science at the University of Tokyo and the Center for Genomic Medicine at the RIKEN Yokohama Institute (BBJ), the Commission of Medical Ethics of the Leuven University Hospital (LEUVEN), the Human Subject Committee of Harvard School of Public Health and the Institutional Review Board of Brigham and Women's Hospital (NHS2 and WGHS), the Ethical Committee of the University of Niigata and the affiliated hospitals (Adachi), the Ethical and Independent Review Services (http://www.eandireview.com) (23andMe), the Danish Research Ethical Committee System (iPSYCH), and the Data Protection Commission of Iceland and the National Bioethics Committee of Iceland (deCODE).

**Genome-wide imputation.** Genotype data within each case–control data set were subjected to sample and SNP quality control as described in the Supplementary Information. Following a shared protocol, each GWA case–control data set was imputed separately. Imputation was carried out using either minimac[58,59] (QIMRHCS, LEUVEN, OX, 23andMe, NHS2-dbGaP, BBJ, WGHS and Adachi-500 k), IMPUTE2 (ref. 60) (iPSYCH) or in-house methods[9] (deCODE). For QIMRHCS, LEUVEN, OX, NHS2-dbGaP, BBJ, WGHS, Adachi-500 K and iPSYCH samples, 1000 Genomes Project March 2012 haplotypes were used as the reference panel, whereas for 23andMe and deCODE samples, 1000 Genomes Project October 2010 haplotypes and whole genome sequence data ($\sim$30 million sequence variants) of 8,453 Icelanders were used as the reference for imputation, respectively. For the Adachi-6 data set, individual-level genotype data for all samples were not available and hence no imputation was performed.

**Genome-wide association analysis.** Imputed genotypes with low imputation quality ($<0.3$ for minimac and $<0.4$ for IMPUTE2) in each data set were excluded from the downstream analysis. Association analysis of the imputed (dosage scores or best guess genotypes) or observed genotypes in each case-control data set including all endometriosis cases ('All') was performed using PLINK[61], SNPTEST[62] or ProbABEL[63], assuming an additive model of genetic inheritance. Imputed genotypes in the 23andMe data set were analysed by adjusting for age and the top five principal components, whereas QIMRHCS, LEUVEN, OX, NHS2-dbGaP, BBJ and Adachi-500 K data sets were analysed without any covariates. For Adachi-6 data set, association analysis was performed using the directly measured genotypes without any covariates. Association analysis of deCODE data was performed using logistic regression[9], and included adjustment for age and population substructures. The iPSYCH were analysed by adjusting for genotyping waves and the top five principal components.

**Genome-wide meta-analysis.** The primary meta-analysis of 'All' endometriosis cases versus controls in the 11 individual case–control data sets was performed using the inverse variance-weighted fixed-effect model in METAL[13]. The $P$ value threshold of $5 \times 10^{-8}$ was declared as genome-wide significant, and SNPs with association at $P < 1 \times 10^{-5}$ were considered to show a suggestive association. Heterogeneity of allelic associations was examined using the Cochran's Q statistic[64] $P < 0.05$, as well as the $I^2$ index[65], which indicates the proportion of variance attributable to between-study heterogeneity. Meta-analysis of SNPs associated in the fixed-effect model at $P < 1 \times 10^{-5}$ and showing evidence of heterogeneity ($P < 0.05$) was also carried out using the Han Eskin random-effects model (RE2)[17] implemented in METASOFT program. Compared to the conventional random-effects model, the RE2 model offers greater power under heterogeneity. The RE2 model relaxes the conservative assumption in hypothesis testing in the traditional RE approach and assumes no heterogeneity under the null hypothesis.

Considering the relatively greater genetic loading (liability) of moderate-to-severe (Grade B) endometriosis (rAFS stage III or IV disease) compared to minimal or mild (Grade A) endometriosis (rAFS stage I or II disease or limited ovarian involvement)[7,8,57], a secondary analysis was performed for SNPs associated at $P < 0.05$, where we performed meta-analysis of Grade B cases versus controls in the QIMRHCS, deCODE, LEUVEN and OX case–control data sets, where disease stage information was readily available.

Per-study Q–Q plots of GWA $P$ values are provided in Supplementary Figs 1–10 as well as the Q–Q plot for GWA meta-analysis $P$ values (Supplementary Fig. 11). We also provide Q–Q plots for the GWA meta-analysis $P$ values, after excluding the eight previously identified risk loci (Supplementary Fig. 12).

**Conditional analysis.** We used GCTA[18] to perform approximate conditional GWA analysis of the newly identified and confirmed risk loci for endometriosis. GCTA allows performing conditional analysis using summary results from GWA meta-analysis and estimated LD from a sufficiently large reference population used in the meta-analysis. Given that the GWA meta-analysis included individuals of both European ($\sim$98%) and Japanese ($\sim$2%) ancestries, we used a reference population with similar proportion of European and Japanese individuals to calculate the LD. Best guess genotypes of well-imputed (minimac $r^2 > 0.8$) SNPs in the QIMRHCS 1,000 Genomes imputed data were then subjected to quality control to exclude SNPs with Hardy-Weinberg Equilibrium $P < 1 \times 10^{-6}$ in controls, MAF $< 0.01$ and $> 5\%$ missingness. As recommended by Yang et al.[66], samples with estimated relationship score $> 0.025$ were also excluded (as opposed to 0.2 used in the meta-analysis), leaving to total of 4,695 samples for further analysis. We then combined the QIMRHCS data set with genotypes of 96 individuals ($\sim$2%) obtained from the 1000 Genomes Japanese reference data, resulting in 4,791 samples with 7,346,981 autosomal SNPs for LD calculation. To examine if the results were influenced by cross-ancestry LD patterns, we also performed additional conditional analysis using summary statistics from European samples alone and calculated LD from only QIMRHCS samples, and compared the results with those from using both QIMRHCS and the 1000 Genomes Japanese samples.

For each genomic locus (new or confirmed) with $P < 5 \times 10^{-8}$, to ensure potential long-range genetic influences were assessed, we conservatively searched $\pm 500$ kb surrounding the lead SNP and adjusted GWA summary data for the lead SNP using the—massoc-cond option in the GCTA[18]. On the basis of genotype data of the reference samples, we first estimated the effective number of independent SNPs within each locus using Genetic type 1 error calculator (GEC)[67], and then totalled them across all the loci examined for secondary signals. Based on the combined QIMRHCS and 1000 Genomes Japanese reference samples, the effective number of SNPs across all the 14 risk loci was 11,631, and hence we declared secondary signal if a SNP achieved $P < 4.3 \times 10^{-6}$ (0.05/11,631).

Where an additional SNP reached threshold for secondary signal after adjustment for the lead SNP, we performed an additional round including both SNPs. If the remaining SNPs had a $P$ value larger than the threshold for secondary signal, no further analysis was performed.

**Characterization of endometriosis-associated SNP effects.** To examine the effect of potential sources of heterogeneity between groups, we compared effect estimates of our 19 independent risk SNPs for Europeans versus Japanese. To address the effects of disease definition, we also compared the effect estimates of studies with a surgically confirmed diagnosis of endometriosis with those with self-reported endometriosis. A Bonferroni-corrected $P < 2.6 \times 10^{-3}$ (corrected for 19 tests) was used to assess statistical significance. Heterogeneity of allelic associations was examined using the Cochran's Q statistic[64] and evidence of heterogeneity was declared if $P_{het} < 0.05$.

**Comparison of identified loci with other traits.** We searched the NHGRI GWAS catalogue[20] for SNP-trait associations, in particular reproductive traits, at our risk loci. SNPs within 500 kb and in LD ($r^2 > 0.7$; arbitrary number) with the lead SNP at each associated locus were identified using the 1000 Genomes Project pilot 1 genotype data and LD values from CEU population. All the SNPs within each locus were then searched in the NHGRI catalogue (downloaded on 3 March 2016) for genome-wide significant associations with other traits or diseases. Using LD-hub[26], we also performed LD score regression analysis[25] to estimate genetic correlation between endometriosis and 199 other traits using summary statistics from fixed-effect meta-analysis including all and Grade B endometriosis cases. A Bonferroni-corrected $P < 2.5 \times 10^{-4}$ (corrected for 199 tests) was used to assess statistical significance.

**Gene-based association analysis.** Gene-based approaches can be more powerful than single SNP analyses, in part due to accounting for allelic heterogeneity (if present) and LD between SNPs, and restricting to genic regions thereby reducing the multiple-testing problem of traditional GWA study. Therefore, using the GWA data from all of the 11 individual case–control data sets, we performed genome-wide gene-based association analysis using VEGAS2 (ref. 28). We first extracted the 4,699,992 SNPs present in all the GWA data sets except Adachi-6 as it includes only observed SNPs and $P$ values from GWA data sets including individuals with European ancestry and the $P$ values from the GWA data sets

including individuals of Japanese ancestry, and analysed separately using VEGAS2. For each gene, the VEGAS test produces a gene-based *P* value by incorporating evidence of association from all SNPs across the gene, while accounting for gene size and LD between SNPs. The resulting gene-based *P* values from GWA studies of European and Japanese ancestry were combined using Stouffer's *Z*-score combined *P* value method. We tested a total of 22,406 genes (including 10 kb 5′ and 3′ to their transcriptional start and end positions) with association results for at least two SNPs, and used a Bonferroni-adjusted significance threshold of $P < 2.23 \times 10^{-6}$ (0.05/22,406) to declare genome-wide significance for gene-based tests.

**Fine-mapping analysis.** For each independent SNP, we defined a genomic region 500 kb on either side of the index SNP and computed 99% credible intervals likely to contain the causal variant using a Bayesian approach, with the strength of evidence for association measured using the Bayes factor for each SNP[68,69]. To assess the resolution of fine-mapping offered by meta-analysis including individuals of both European and Japanese ancestries, we calculated the 99% credible sets based on GWA meta-analysis results including all studies but Adachi-6, as well as results based on only European studies. We did not compute the 99% credible sets for the Japanese alone because the small sample size makes comparison of fine-mapping intervals meaningless. We calculated approximate Bayes' factor (BF) for each SNP using:

$$\mathrm{BF}_i = \frac{\sqrt{1 - R_i}}{\exp\left(-\frac{R\beta_i^2}{2\sigma_i^2}\right)}$$

where $\beta_i$ is the allelic effect of the *i*th SNP, with corresponding standard error $\sigma_i$, and $R_i = 0.04/(\sigma_i^2 + 0.04)$, which incorporates a $N(0, 0.2^2)$ prior for $\beta_i$ assigning high probability to small effect sizes and only small probability to large effect sizes. Adachi-6 data set with only observed SNPs was excluded to maintain uniform SNP coverage across studies. Using the below formula, we calculated the posterior probability that the *i*th SNP is causal:

$$\varphi_i = \frac{\mathrm{BF}_i}{\sum_i \mathrm{BF}_i}$$

where the summation in the denominator is over all the SNPs passing quality control across the locus. We assumed a single causal variant for each association signal, calculated BF for all SNPs within 500 kb of the index SNP, and ranked the SNPs based on their BF. We then constructed a 99% credible set of variants by combining the ranked SNPs, and calculated the number of SNPs and length of genomic region covered by each credible set.

**Identification of putative functional variants.** We examined the *cis* associations between the 19 independent SNPs and other SNPs in high LD ($r^2 > 0.7$) with the lead SNPs and expression of nearby genes using the GTEx eQTL portal[29]. A total of 395 SNPs were searched.

We used summary data-based Mendelian randomization (SMR)[30] to identify potentially causal genes underlying the identified endometriosis associations. Briefly, the SMR method exploits the concept of MR analysis; it tests for the causative effect of an exposure on an outcome using a genetic variant (for example, SNP) as an instrumental variable. In principle, it uses the MR analysis (for example, two-stage least squares approach) to search for causal genes at the loci identified from GWA studies for complex traits. Using summary data from GWA and eQTL studies, the SMR method tests the association between a trait and the expression level of each gene across the whole genome. We used the method to identify causal genes at our endometriosis loci, using the GWA meta-analysis summary results from all studies including all endometriosis cases, and the eQTL summary data from Westra et al.[31], an eQTL meta-analysis of 5,311 samples from peripheral blood, with SNPs imputed to the HapMap2 reference panels. Of the 14,329 probes in the eQTL data, only probes with at least one *cis*-eQTL at $P < 5 \times 10^{-8}$ were included while excluding probes in the MHC region, resulting in 5,967 probes for final SMR analysis. Therefore, genes with $P < 8.4 \times 10^{-6}$ (equivalent to 0.05/5,967) were declared to achieve genome-wide significance in the SMR analysis. SMR method also tests this colocalization using the HEIDI (Heterogeneity In Dependent Instruments) test. A $P < 0.05$ in the HEIDI test suggests that the majority of the associations identified by the SMR test could be explained by colocalization.

**Pathway analysis.** We used Data-driven Expression-Prioritized Integration for Complex Traits (DEPICT)[32] to identify genes and pathways responsible for the observed genetic associations, thereby gaining biological insights at the identified risk loci. Using comprehensive data on gene expression, molecular pathways, experimentally derived protein-protein interactions, phenotypic gene sets, Reactome and KEGG pathways and gene ontology terms, DEPICT highlights the causal genes at each risk locus, enriched pathways and the relevant tissues/cell types where associated genes are highly expressed. Based on the results from the fixed-effect meta-analysis of all the GWA data set, we ran DEPICT analyses on (i) SNPs showing genome-wide significant ($P < 5 \times 10^{-8}$) association, and (ii) SNPs with suggestive ($P < 1 \times 10^{-5}$) association signal in GWA meta-analysis including all or Grade B cases.

**Variance explained.** Based on Neil Risch's liability threshold model[19], we estimated proportion of variance explained by a single SNP using the effect allele frequency and odds ratio from the GWA meta-analysis of European studies. We used population prevalence of 8 (refs 7,70) and 2.5% (ref. 71) for all and Grade B endometriosis cases, respectively.

Assuming associated SNPs are not in high LD, we calculated the sum of single-SNP explained variances to produce the total variance explained by a set of independent SNPs[72].

**Data availability.** The authors declare that the data supporting the findings of this study are available within the article and its Supplementary Information Files. For additional data (beyond those included in the main text and Supplementary Information) that support the findings of this study, please contact the corresponding authors.

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

## Acknowledgements

We acknowledge all the study participants in 11 individual endometriosis studies that provided an opportunity for the current study. We also thank many hospital directors and staff, gynaecologists, general practitioners and pathology services in Australia who provided assistance with confirmation of diagnoses. We would like to thank the research participants and employees of 23andMe for making this work possible. We thank the subjects of the Icelandic deCODE study for their participation. We thank research staff and clinicians for providing diagnostic confirmation for the OX data set. We would like to express our gratitude to the staff and members of the Biobank Japan and Laboratory for Statistical Analysis, RIKEN Center for Integrative Medical Sciences for their outstanding assistance. The QIMR study was supported by grants from the National Health and Medical Research Council (NHMRC) of Australia (241,944, 339,462, 389,927, 389,875, 389,891, 389,892, 389,938, 443,036, 442,915, 442,981, 496,610, 496,739, 552,485, 552,498, 1,026,033 and 1,050,208), the Cooperative Research Centre for Discovery of Genes for Common Human Diseases (CRC), Cerylid Biosciences (Melbourne) and donations from N. Hawkins and S. Hawkins. Analyses of the QIMRHCS and OX GWAS were supported by the Wellcome Trust (WT084766/Z/08/Z) and makes use of WTCCC2 control data generated by the Wellcome Trust Case-Control Consortium (awards 076113 and 085475). The iPSYCH study was funded by The Lundbeck Foundation, Denmark (R102-A9118, R155-2014-1724 ), and the research has been conducted using the Danish National Biobank resource supported by the Novo Nordisk Foundation. A full list of the investigators who contributed to the generation of these data is available from http://www.wtccc.org.uk. D.R.N. was supported in part by the NHMRC Fellowship (613674) and ARC Future Fellowship (FT0991022) schemes. E.G.H. (631096) and G.W.M. (339446, 619667) were supported by the NHMRC Fellowships Scheme. S.M. is supported by an Australian Research Council Future Fellowship. A.P.M. was supported by a Wellcome Trust Senior Research Fellowship (award WT098017). N.R. was supported by funding from the Medical Research Council UK (MR/K011480/1). This study was funded by the BioBank Japan project, which is supported by the Ministry of Education, Culture, Sports, Sciences and Technology of Japanese government.

## Author contributions

S.K.L., K.T.Z., S.A.M., T.D., D.I.C., K.S., J.Y.T., G.W.M. and D.R.N. conceived and supervised the study. Y.S., A.P.M., G.W.M. and D.R.N. designed analytical strategies. Y.S., A.P.M., A.J.S., S.M., G.T., D.I.C., G.W.M. and D.R.N. performed genome-wide association analysis and imputation. Y.S. performed GCTA conditional analysis, characterization of SNP loci, cross-trait analysis, gene-based analysis, Bayesian

fine-mapping analysis, pathway and bioinformatic analyses and GWA meta-analyses. F.Z. and J.Y. performed SMR analysis. V.S., A.F., I.D.V., J.E.B., T.L.E., K.M.R., P.M.R., N.G.M., S.A., M.K., L.M.W., D.R.V.E., M.N., K.T.Z., S.A.M., T.D., G.W.M., D.I.C., K.S., J.Y.T. and D.R.N. contributed to GWAS data, sample preparation and clinical phenotyping. A.P.M., L.M.W., D.R.V.E., M.N., K.T.Z., S.A.M., T.D., G.W.M., D.I.C., K.S., J.Y.T. and D.R.N. conducted genotyping and quality control of the data. A.P.M., I.D.V., S.M., N.G.M., D.R.V.E., M.N., K.T.Z., S.A.M., T.D., G.W.M., D.I.C., K.S., J.Y.T. and D.R.N. obtained study funding. Y.S. drafted the manuscript. Y.S., V.S., A.P.M., A.F., N.R., I.D.V., J.E.B., F.Z., T.L.E., S.J., D.O., D.P., K.M.R., P.M.R., A.J.S., S.M., N.G.M., C.M.B., S.A., K.Y., T.E., A.T., Y.K., K.M., M.K., G.T., R.T.G., U.T., L.M.W., J.Y., D.R.V.E., M.N., S.-K.L., K.T.Z., S.A.M., T.D., G.W.M., D.I.C., K.S., J.Y.T. and D.R.N. revised the manuscript and provided final approval.

## Additional information

**Competing interests:** V.S., G.T., U.T. and K.S. are employees of the biotechnology firm deCODE Genetics, a subsidiary of AMGEN. The remaining authors declare no competing financial interests.

## iPSYCH-SSI-Broad group

Thomas M. Werge[24,28,29] and Wesley K. Thompson[24,28,30]

[28]Institute of Biological Psychiatry, Mental Health Center, Sct. Hans, Mental Health Services, DK-2100 Copenhagen, Denmark; [29]Institute of Clinical Sciences, Faculty of Medicine and Health Sciences, University of Copenhagen, DK-2200 Copenhagen, Denmark; [30]Department of Psychiatry, University of California, San Diego, La Jolla, California 92093, USA.

