## [Peer Review File · Nature Communications]

Reviewer #1 (Remarks to the Author):

The manuscript is an extension of previous GWAS studies of endometriosis, which has increased the sample size considerably and found five new loci. The paper has used robust methodology with appropriate controls and while not made significant advances in understanding the biology or treatment of endometriosis, this is a step forward in understanding. In particular the use of recently published methods to try and determine causative variants is useful. It is also interesting to note the difference in variance explained by the GWAS loci depending on the phenotypic definition used, which has potential impact for other traits and diseases.

Reviewer #2 (Remarks to the Author):

The manuscript by Sapkota et al describes the largest assessment of the genetic basis of endometriosis to date. Many of the analyses appear to be thoroughly conducted and relevant. It was unfortunate, and perhaps surprising, that given the impressive sample size more novel loci were not discovered. I have a number of comments for consideration:

1) The introduction states 11 known signals in European ancestry samples – results section (line 164) begins by confirming the “eight previously identified” signals are still GWS. Throughout the paper the authors complicate the description of known or novel loci, particularly with reference to the KDR signal. I would suggest it would be much simpler to stick with 11 known (or 12 if including the Japanese locus) and describe the findings with reference to that. I assume the other 2 loci were not significant? Much of the results section could be streamlined when describing the status and identification of known/novel loci.

2) the “Endometriosis risk SNPs associated with other traits” section lists pleiotropic associations in an uninformative way and could be summarised in a supplementary table. I’d recommend that the authors should instead be systematic by performing whole genome genetic correlations (for example, using LD hub) and follow up with summary based mendelian randomisation analyses where appropriate. This could prove useful for the breast cancer link – is there a dose response relationship there?

3) It makes little sense to me to perform GTEX expression lookup for 1167 SNPs rather than the 19 index SNPs (if I’m right in saying that has been done). This analysis should be repeated and coincidental overlap with expression signals (i.e if the endometriosis/expression hits are in low LD) excluded.

4) bioinformatic analysis of risk loci section – first paragraph uninformative and should be cut down or removed. Is it interesting that 76 SNPs have a CADD score ≥ 10 ? I’d imagine that any trait associated variant in the genome had a greater chance of being in this category.

5) Could the smaller effect estimate in any case endometriosis compared to grade B relate simply to disease misclassification? This should be discussed.

6) I felt there was too much emphasis on heterogeneity between studies rather than heterogeneity between clinical reporting strategy and disease severity. Study names are used frequently throughout the text which is uninformative – instead focus should be on the sources of detected heterogeneity. Forrest plots could be updated to reflect this.

7) Last sentence in abstract and parts of discussion overstate findings. Many other studies (not limited to GWAS) describe the relevance of steroid hormones in endometriosis. The findings in this current study provide no additional evidence for an extensive role of this mechanism (besides 2-3 potentially functional genes) – pathway analyses did not show enrichment of the many other genes involved in sex steroid hormone signalling.

8) It would be useful for endometriosis researchers (and the broader scientific community) if the meta-analysis was made publicly available after publication. This is increasingly common for other comparable complex traits. Can the authors commit to this? (I understand data from sources such as 23andMe might need to be excluded, but the remaining data is still valuable).

Minor:

- Some supplementary tables were incorrectly formatted during PDF conversion and contain missing values (i.e lots of Table S4 contains "####"). This made it hard to appropriately evaluate them.

- Introduction should mention reference panels used in each of the studies (i.e All Hapmap 2 or 1KG or exome...)

- rsnumbers uninformative in introduction and elsewhere – stick to gene names or cytobands

- consistent gene names to label loci should be used throughout (with appropriate disclaimer on labelling) – e.g ESR1 is referred to as CCDC170 in table 1.

- Table 1 – Useful to add sample size for grade B analysis. It would also be useful to add all 14 lead loci in this table, split by novelty.

- line 218 – “the ten new SNPs identified in this study ...” – all ten haven’t been described at this stage

- Line 396 - not sure $FDR < 0.2$ constitutes strong evidence for the ten enriched tissue types

Reviewer #3 (Remarks to the Author):

In this study the authors performed a meta-analysis of previous GWAs increasing the sample size to a total of 208,000 individuals. This is particularly important as it is known that a large sample size allows to detect association of SNPs with small effects and allows to partially overcome some limitations of phenotypic characterization especially in controls. The statistical analysis is appropriate and, in my opinion, leads to convincing results. The new loci identified harbour candidate genes involved in the sex steroid pathways and some of them are also linked to other sex hormones dependent pathologies. This is not unexpected given the role of steroid hormones in the pathogenesis of endometriosis but can pave the way to more targeted functional researches thus confirming the main goal of GWAs.

As a minor suggestion, I would consider to delete some of the supplementary figures and tables that could be redundant.

We thank the reviewers for their positive and helpful comments on our manuscript. We have addressed all comments with clarifications in the manuscript. Below is a detailed point-by-point response.

Reviewer #1 (Remarks to the Author):

The manuscript is an extension of previous GWAS studies of endometriosis, which has increased the sample size considerably and found five new loci. The paper has used robust methodology with appropriate controls and while not made significant advances in understanding the biology or treatment of endometriosis, this is a step forward in understanding. In particular the use of recently published methods to try and determine causative variants is useful. It is also interesting to note the difference in variance explained by the GWAS loci depending on the phenotypic definition used, which has potential impact for other traits and diseases.

Thank you.

Reviewer #2 (Remarks to the Author):

The manuscript by Sapkota et al describes the largest assessment of the genetic basis of endometriosis to date. Many of the analyses appear to be thoroughly conducted and relevant. It was unfortunate, and perhaps surprising, that given the impressive sample size more novel loci were not discovered. I have a number of comments for consideration:

1) The introduction states 11 known signals in European ancestry samples – results section (line 164) begins by confirming the “eight previously identified” signals are still GWS. Throughout the paper the authors complicate the description of known or novel loci, particularly with reference to the *KDR* signal. I would suggest it would be much simpler to stick with 11 known (or 12 if including the Japanese locus) and describe the findings with reference to that. I assume the other 2 loci were not significant? Much of the results section could be streamlined when describing the status and identification of known/novel loci.

We agree with the reviewer that there could be more clarity with respect to the number of previously reported and novel loci. We have reorganized this in the revised version, and also addressed the confusion with the *KDR* signal. We have updated Tables and Figures to include the *KDR* signal as a known locus. Throughout the manuscript, we have now adhered to the 11 previously reported European SNP risk loci, and described the results with respect to that. Of these, we confirmed 9 loci and we could not replicate associations at the remaining two.

2) the “Endometriosis risk SNPs associated with other traits” section lists pleiotropic associations in an uninformative way and could be summarised in a supplementary table. I’d recommend that the authors should instead be systematic by performing whole genome genetic correlations (for example, using LD hub) and

follow up with summary based mendelian randomisation analyses where appropriate. This could prove useful for the breast cancer link – is there a dose response relationship there?

Thank you. We have significantly revised the first paragraph in “**Endometriosis risk SNPs associated with other traits**” section and it reads as below:

*We checked genome-wide significant associations between the 19 SNPs and other diseases or traits using the NHGRI GWAS catalog²⁰. We searched for 395 SNPs including the 19 SNPs as well as all other SNPs in high LD ($r^2 > 0.7$) with the 19 SNPs (**Supplementary Table 7**). Of these, we observed associations with multiple diseases or traits, including epithelial ovarian cancer²¹, low-density lipoprotein cholesterol²², coronary heart disease²³, luteinizing hormone and follicle-stimulating hormone levels, and age at onset for menopause²⁴.*

As suggested by the reviewer, we also performed LD score regression analysis using LD hub to examine genetic correlation between endometriosis and other traits. We performed this analysis for both all and Grade B endometriosis cases. Of the 199 traits available in the LD hub database, no trait showed significant genetic correlation with endometriosis ($P < 2.5 \times 10^{-4}$ after multiple testing). We have provided results with $P < 0.05$ in **Supplementary Figures 43 and 44**. To reflect this analysis, we have added the paragraphs below in Methods and Results sections:

Methods (Comparison of identified loci with other traits):

Using LD-hub, we also performed LD score regression analysis to estimate genetic correlation between endometriosis and 199 other traits using summary statistics from fixed-effect meta-analysis including all and Grade B endometriosis cases. A Bonferroni-corrected $P < 2.5 \times 10^{-4}$ (corrected for 199 tests) was used to assess statistical significance.

Results (Endometriosis risk SNPs associated with other traits)

Using LD-hub²⁶, we also performed LD score regression analysis²⁵ to estimate genetic correlation between endometriosis and 199 other traits using summary statistics from fixed-effect meta-analysis including all and Grade B endometriosis cases. A Bonferroni-corrected $P < 2.5 \times 10^{-4}$ (corrected for 199 tests) was used to assess statistical significance.

3) It makes little sense to me to perform GTEX expression lookup for 1167 SNPs rather than the 19 index SNPs (if I'm right in saying that has been done). This analysis should be repeated and coincidental overlap with expression signals (i.e if the endometriosis/expression hits are in low LD) excluded.

We agree the list of 1167 SNPs have not been screened for LD with the index SNPs. We updated the analysis by examining the 19 index SNPs and other SNPs that are in high LD ($r^2 > 0.7$) with the index SNPs. We have revised both Methods and Results section to reflect this, and reads as below in the Results section:

*We then examined the cis associations between the 19 independent SNPs and other SNPs in high LD ($r^2 > 0.7$) with the lead SNPs (**Supplementary Table 7**).*

4) bioinformatic analysis of risk loci section – first paragraph uninformative and should be cut down or removed. Is it interesting that 76 SNPs have a CADD score ≥ 10 ? I'd imagine that any trait associated variant in the genome had a greater chance of being in this category.

We agree with the reviewer and have removed the first paragraph, as well as the corresponding text in Methods.

5) Could the smaller effect estimate in any case endometriosis compared to grade B relate simply to disease misclassification? This should be discussed.

We have added the below paragraph in the Discussion.

The majority of the identified SNP loci showed larger effects in the Grade B analysis in comparison to those based on analysis of all endometriosis. This observation is consistent with the previous reports of greater genetic loading in moderate-to-severe endometriosis^{7,8,58}, and the genetic burden generally increases from less severe (minimal) to more severe disease, consistent with disease progression. The smaller effect sizes in analysis including all endometriosis cases may be due to disease misclassification in self-reported endometriosis cases. However, this would only be part of the explanation. We have previously shown the genetic contribution to patients with surgically confirmed endometriosis and minimal disease is weaker than for patients with severe disease^{7,58}. Using polygenic prediction analysis, we also showed significant prediction for minimal disease between two independent datasets with surgically confirmed endometriosis⁵⁸.

6) I felt there was too much emphasis on heterogeneity between studies rather than heterogeneity between clinical reporting strategy and disease severity. Study names are used frequently throughout the text which is uninformative – instead focus should be on the sources of detected heterogeneity. Forrest plots could be updated to reflect this.

We have updated paragraphs describing our analysis underlying the observed between-study heterogeneity, and they read as below:

*Of the 14 distinct loci, six [2p14, 2q35, 4q12, 6q25.1 (SYNE1), 7p15.2, and 12q22] showed evidence of between-study heterogeneity ($P_{het} < 0.05$; I^2 : 13.30-20.33) in fixed-effect meta-analysis including all endometriosis cases; however, after appropriately modelling the observed heterogeneity in the RE2 model, these associations remained genome-wide significant (**Supplementary Table 2**).*

We next examined SNP effects with respect to disease severity, ancestry and diagnosis

of endometriosis to investigate potential sources of observed between-study heterogeneity (Supplementary Tables 2 and 6). None of the six loci showed between-study heterogeneity in analyses restricted to Japanese alone and self-reported endometriosis cases (Supplementary Table 6). In Europeans, five [2p14, 2q35, 4q12, 6q25.1 (SYNE1) and 7p15.2] showed heterogeneity in allelic associations, and except 6q25.1 (SYNE1), this trend persisted in surgically confirmed endometriosis cases. The between-study heterogeneity at 12q22 vanished in Europeans alone but was again apparent in surgically confirmed endometriosis cases. Interestingly, except for 2p14 and 4q12, heterogeneity in SNP associations at the remaining four loci attenuated in Grade B analysis restricting to only surgically confirmed moderate-to-severe endometriosis cases (Supplementary Table 2). Additionally, all five loci but 2p14 produced larger effect sizes for surgically confirmed endometriosis than diagnosis based on self-reports, and 2p14 produced larger effect size in Japanese than Europeans. With respect to 2p14, residual between-study heterogeneity may be driven by opposite direction of effect in two out of the eight European studies (Supplementary Table 2 and Figure 26).

We have also updated the **Supplementary Table 6** by including heterogeneity test P values.

Wherever possible, we also removed study names that were not informative.

7) Last sentence in abstract and parts of discussion overstate findings. Many other studies (not limited to GWAS) describe the relevance of steroid hormones in endometriosis. The findings in this current study provide no additional evidence for an extensive role of this mechanism (besides 2-3 potentially functional genes) – pathway analyses did not show enrichment of the many other genes involved in sex steroid hormone signalling.

The novel variants near *ESR1* and *FSH* provide important evidence for specific parts of the pathway that may be altered in endometriosis. However, we agree with the reviewer that many other studies support a role for steroid hormones in endometriosis and have modified the last sentence of the abstract and discussion.

These results identify novel variants in or near specific genes with important roles in sex steroid hormone signalling and function, and offer unique opportunities for more targeted functional research efforts.

8) It would be useful for endometriosis researchers (and the broader scientific community) if the meta-analysis was made publicly available after publication. This is increasingly common for other comparable complex traits. Can the authors commit to this? (I understand data from sources such as 23andMe might need to be excluded, but the remaining data is still valuable).

We always strive to make our results available to other researchers. Results from the 23andMe data can only be shared for the top 10,000 SNPs, and we are currently

pursuing approvals from each dataset to centralize approval and facilitate the sharing of our results. Furthermore, we have added the following statement under “Data availability” at the end of our manuscript (page 27 of 38): “*The authors declare that the data supporting the findings of this study are available within the article, its supplementary information files and on request.*”

Minor:

- Some supplementary tables were incorrectly formatted during PDF conversion and contain missing values (i.e lots of Table S4 contains “####”). This made it hard to appropriately evaluate them.

We apologize for the inconvenience, and have now ensured the Table is shown clear for reviewer’s evaluation.

- Introduction should mention reference panels used in each of the studies (i.e All Hapmap 2 or 1KG or exome...)

Of the 11 individual studies, ten studies are imputed up to a recent 1000 Genomes Project reference panel and one study consists of observed directly genotyped data alone. This information is now included in the Introduction section.

- rsnumbers uninformative in introduction and elsewhere – stick to gene names or cytobands

We have removed rsIDs and used the cytobands throughout the text, where possible. However, we have retained the rsIDs and cytobands at their first mention in the introduction and results. We also retain rsIDs (and nearest gene information) in some sections of text to assist the reader interpret findings where multiple independent SNP risk loci are implicated at the same gene/cytoband locus. For example, rs10965235 in *CDKN2BAS* on chromosome 9p21.3 and rs1537377 near *CDKN2B-AS1* on 9p21.3, and at the 5 loci listed in Table 2.

-consistent gene names to label loci should be used throughout (with appropriate disclaimer on labelling) – e.g ESR1 is referred to as *CCDC170* in table 1.

Table 1 (updated) consists of results of the 14 distinct loci, including 2 at 6q25.1 (*CCDC170* and *SYNE1*). Conditional analysis identified secondary signal also at 6q25.1 but in *ESR1*.

-Table 1 – Useful to add sample size for grade B analysis. It would also be useful to add all 14 lead loci in this table, split by novelty.

We have combined results for the 14 loci in one table (Table 1), and updated other table numbers accordingly.

-line 218 – “the ten new SNPs identified in this study ...” – all ten haven’t been described at this stage

We moved the paragraph containing this phrase after the GCTA conditional analysis, which identified further five independent SNPs, totaling ten novel SNP associations.

- Line 396 - not sure $FDR < 0.2$ constitutes strong evidence for the ten enriched tissue types

We have modified the sentence and now it reads as below:

Additionally, suggestive evidence for enrichment ($FDR \leq 0.2$) was observed for ten tissues, including female genitalia, uterus, endocrine glands, endometrium, ovary and Fallopian tubes (Supplementary Table 16).

Reviewer #3 (Remarks to the Author):

In this study the authors performed a meta-analysis of previous GWAs increasing the sample size to a total of 208,000 individuals. This is particularly important as it is known that a large sample size allows to detect association of SNPs with small effects and allows to partially overcome some limitations of phenotypic characterization especially in controls. The statistical analysis is appropriate and, in my opinion, leads to convincing results. The new loci identified harbour candidate genes involved in the sex steroid pathways and some of them are also linked to other sex hormones dependent pathologies. This is not unexpected given the role of steroid hormones in the pathogenesis of endometriosis but can pave the way to more targeted functional researches thus confirming the main goal of GWAs. As a minor suggestion, I would consider to delete some of the supplementary figures and tables that could be redundant.

Thank you. We have removed Supplementary figures containing LocusZoom plots for endometriosis risk loci, which also harbored SNPs associated with other traits. This information is summarized in Supplementary Table 9 as well, as correctly pointed by the reviewer.

Reviewer #2 (Remarks to the Author):

I am satisfied with the responses to my comments.